# Low rate of SARS-CoV-2 incident infection identified by weekly screening PCR in a prospective year-long cohort study

**Whitney E. Harrington[1,2], Winnie Yeung[1], Ingrid A. Beck[1], Fred D. Mast[1], John Houck[1], Sheila Styrchak[1], Leslie R. Miller[1], Song Li[1], Micaela Haglund[1], Yonghou Jiang[1], Blair Armistead[1], Jackson Wallner[1], Tina Nguyen[1], Daisy Ko[1], Samantha Hardy[1], Alyssa Oldroyd[1], Ana Gervassi[1], John D. Aitchison[1,2,3], Lisa M. Frenkel[1,2]\***

1 Center for Global Infectious Disease Research, Seattle Children's Research Institute, Seattle, Washington, United States of America, 2 Department of Pediatrics, University of Washington, Seattle, Washington, United States of America, 3 Department of Biochemistry, University of Washington, Seattle, Washington, United States of America

\* lfrenkel@uw.edu

**Data Availability Statement:** All relevant data are within the paper and its Supporting Information files.

**Funding:** Seattle Children's Research Institute [to LMF, WEH, and JDA]; the National Institute of

## Abstract

### Background

Asymptomatic and pre-symptomatic SARS-CoV-2 infections may contribute to ongoing community transmission, however, the benefit of routine screening of asymptomatic individuals in low-risk populations is unclear.

### Methods

To identify SARS-CoV-2 infections 553 seronegative individuals were prospectively followed for 52 weeks. From 4/2020-7/2021, participants submitted weekly self-collected nasal swabs for rtPCR and completed symptom and exposure surveys.

### Results

Incident SARS2-CoV-2 infections were identified in 9/553 (1.6%) participants. Comparisons of SARS2-CoV-2(+) to SARS2-CoV-2(-) participants revealed significantly more close contacts outside the household (median: 5 versus 3; p = 0.005). The incidence of infection was higher among unvaccinated/partially vaccinated than among fully vaccinated participants (9/7,679 versus 0/6,845 person-weeks; p = 0.004). At notification of positive test result, eight cases were symptomatic and one pre-symptomatic.

### Conclusions

These data suggest that weekly SARS2-CoV2 surveillance by rtPCR did not efficiently detect pre-symptomatic infections in unvaccinated participants.

Allergy and Infectious Diseases [K08 AI135072 to WEH]; and the Burroughs Wellcome Fund [CAMS 1017213 to WEH]. Support for REDCap, which was utilized for data management, came from Institute of Translational Health Science (ITHS) grant support (UL1 TR002319, KL2 TR002317, and TL1 TR002318). The funders had no role in study design, data collection and analysis, decision to publish, or preparation of the manuscript.

**Competing interests:** The authors have declared that no competing interests exist.

# Background

Limiting interactions with others ("social distancing") by masking and physical spacing, enhanced hand hygiene, and vaccination are modalities shown to reduce SARS-CoV-2 (**SARS2**) transmission. The impact of additional mitigation strategies such as routine screening of asymptomatic individuals is less clear. Periodic testing for SARS2 infection may detect asymptomatic or pre-symptomatic infections, which have been reported to represent a significant proportion of infections particularly amongst healthy populations without comorbidities [1]. Asymptomatic and pre-symptomatic infections may contribute to transmission, albeit less than symptomatic infection [2], and therefore surveillance has been recommended to detect such cases [3]. Weekly testing for SARS2 infection is recommended for United States federal government employees who are unvaccinated when transmission risk is medium or high to reduce work-place transmission [4]. Little is known about the usefulness of this approach in detecting asymptomatic or pre-symptomatic infections, particularly amongst low-risk populations.

We aimed to detect SARS2 infections by weekly real-time PCR (**rtPCR**) testing over a year, as part of a larger study to identify biomarkers of SARS2 disease severity.

# Materials and methods

Individuals were invited to enroll into a prospective, 52 week-long study to detect SARS2 infections between April and July 2020. Eligibility criteria included work-force members from Seattle Children's Research Institute and their household members of any age. Participants were excluded if they had a fever or an acute respiratory or gastrointestinal illness at the time of enrollment. Participants were taught how to self-collect nasal samples during enrollment and written instructions were provided for future reference. The study was approved by the Seattle Children's IRB (STUDY00002434), and all adult participants provided written informed consent. Guardians provided written consent for children, and children ≥7 years of age provided verbal assent.

Participants self-collected nasal samples for SARS2 rtPCR once each week. Nasal swabs were collected from children by their guardians. A polyester swab (Fisherbrand, Cat# 22363170) was inserted a distance of ~1.5cm sequentially in each nostril and rotated against the mucosal surface 8–10 times, then placed into a dry, sterile tube, capped, and returned to the study team. Swabs were tested for SARS2 RNA within 72 hours of collection. Additional swabs could be submitted for testing when participants experienced symptoms or were exposed to a contact with SARS2 infection.

SARS-CoV-2 diagnosis was performed using a laboratory-developed rtPCR assay with Emergency Use Authorization (**EUA**) from the Washington State Department of Health. The rtPCR targeted two regions of the SARS-CoV-2 nucleocapsid gene (N1 and N2) by specific reverse transcription of RNA and amplification (iTaq™ Universal Probes One-Step Kit, Bio-Rad Laboratories, CA) [5]. The analytical sensitivity of the RT-qPCR was 10 SARS-CoV-2 RNA copies/reaction as determined by serial dilutions of an RNA control with the cut-off for SARS-CoV-2 positivity set at a cycle threshold (Ct) of 40. The limit of detection for each participant's nasal specimen was 250 SARS-CoV-2 RNA copies/nasal swab. During the development of the CLIA-certified laboratory-developed assay, testing confirmed that dry swabs had comparable sensitivity to swabs placed in viral transport media or in phosphate buffered saline for up to 72 hours. When a self-collected swab tested positive for both SARS2 targets, the participant was considered to have a presumptive infection, and a second specimen was collected by the study team or health-care worker to confirm SARS2 infection. Confirmed positive tests were reported to the Washington State Department of Health. Clinical SARS2 positive rtPCR

tests obtained by participants primarily at Seattle Children's and University of Washington facilities due to symptoms were voluntarily reported to the study team and considered an infection; outside records were not directly reviewed.

Participants completed weekly surveys to assess social distancing (e.g., enumerating close contacts while unmasked), report contact with SARS2-infected individuals, and record interval symptoms. Close contacts were unmasked interactions within 6 feet for ≥15 minutes. These were tallied separately for contacts within and outside of the participant's household. Queried symptoms included: fever, sore throat, sneezing, cough, vomiting, diarrhea, loss of smell/taste, headache, muscle/body aches, shortness of breath, chest tightness, fatigue/tiredness, and others (free response). Participants who tested SARS2(+) were asked to complete additional daily symptom logs. Symptoms were compared with the National Institute of Health (NIH) COVID-19 Treatment Guidelines Clinical Spectrum guidance to determine severity of infection. Information on vaccine receipt (date, type) was obtained when SARS2 vaccines became available in the United States. Participants' plasma was tested for anti-Spike antibodies using the SCoV-2 DetectTM IgG and IgM ELISA (InBios, Seattle, WA) at enrollment for all participants and at the end of the study if unvaccinated and without prior infection; antibodies against nucleocapsid protein were not assessed.

Continuous demographic variables were compared in participants with and without incident infection using a quantile regression of the median, and distributions of categorical variables were compared with a Fisher's exact test. Statistics were conducted in Stata version 14.

## Results

The study enrolled 558 individuals (**Fig 1**). Antibodies (IgG) to SARS2 Spike were detected at enrollment in five participants, who were excluded from further analysis. At study entry the remaining 553 participants ranged in age from four months to 80 years (median: 36 years); 77 children under the age of 18 were enrolled of whom 27 were under the age of five years. Participants self-reported genders included 292 (54%) females and 261 (46%) males. Participants contributed a median of 48 nasal swabs (range: 0–60) for SARS2 rtPCR and completed a median of 43 weekly surveys (range: 0–52).

During the 52-week study, participants' close contacts were relatively limited, with a median of three close contacts reported within participants' households (range: 0–11), and three close contacts outside of their households (range: 0–28). Similarly, known exposures to an individual positive for SARS2 infection were limited, with 33 individuals (6%) reporting a range of 1–4 total exposures. Vaccines against SARS2 became available during the study and "full vaccination" was reported by 391 (71%) participants, which was defined as fourteen days after two doses of BNT162b2 (Pfizer-BioNTech) received by 216 (39%), two doses of mRNA-1273 (Moderna) by 164 (30%), or one dose of JNJ-78436735 (Janssen) by eight (2%) participants (**Fig 1**), and by two doses of unreported vaccine products by three individuals (1%). In total, we observed 6,419 unvaccinated or vaccine status unknown person weeks, 1,296 partially vaccinated person weeks, and 6,845 fully vaccinated person weeks. Vaccine boosters (i.e., 3$^{rd}$ dose for BNT162b2 or mRNA-1273, or a 2$^{nd}$ dose for JNJ-78436735) were not available during the study.

A total of 24,095 nasal swabs were submitted and tested during the study. SARS2 was detected in 26 swabs with incident SARS2 infections detected in 9/553 participants (1.6%; 95% CI 0.8, 3.1) (**Fig 1**). Participants with incident infections ranged in age from one to 60 years (median 29) and five (56%) were female. Three participants had SARS2 exposures within two weeks of their incident infection, all within their household. During the week prior and week of SARS2 diagnosis, the mean number of close contacts among all nine SARS2-positive

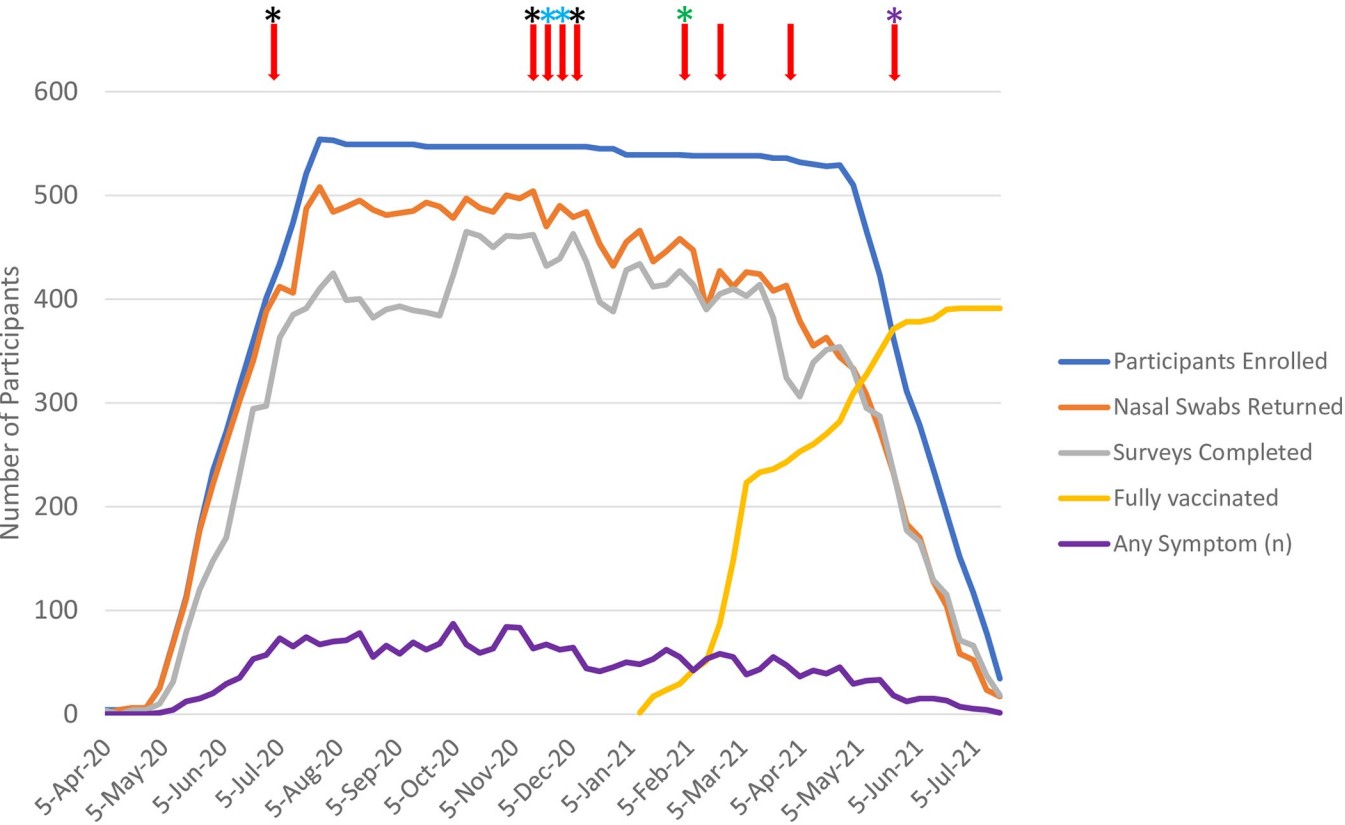

**Fig 1. Number of SARS2 rtPCR tests, weekly surveys, and SARS2 cases by calendar week.** The study was conducted across a total of 6,411 unvaccinated person weeks, 1,268 partially vaccinated person weeks (<14 days from complete vaccination), and 6,845 fully vaccinated weeks (≥14 days from complete vaccination). SARS2 cases denoted with red arrows. Most cases had a discrete risk factor: black star = health care worker; blue star = family member of SARS2 (+) healthcare worker; green star = household SARS2(+) exposure; purple star = attending in person school. No stars = no identified risk factor.

participants was three within and four outside of their households. The infected participants included three healthcare workers, two family members of an infected healthcare worker (healthcare worker was not enrolled in the study), a high-school student attending in-person classes, a parent of a child who was part of a daycare outbreak (child was not enrolled in the study), and two household members of Seattle Children's employees. All nine cases had symptomatic SARS2 with three reporting fever; all were classified as 'Mild Illness' per NIH COVID-19 Treatment Guidelines. Cases were notified of their SARS2(+) results at a median of 1 day (range 1–4) after swab collection, at which time 1/9 (95% CI 0.01, 46) was in the pre-symptomatic phase (notified 2 days after swab collected), a second had mild symptoms mis-attributed to "allergies" (notified 4 four days after swab collected), while the remaining seven had sought testing due to COVID-19 symptoms. Eight cases occurred in unvaccinated participants and one case in a participant 17 days after his first dose of the Pfizer vaccine.

Comparisons of participants with incident SARS2 infections to those in whom infection was not detected across the duration of the study found similar ages (median: 29 versus 36, p = N.S.), gender (56% versus 54% female, p = N.S.), and similar numbers of household contacts (median: 3 versus 3, p = N.S.). However, participants with incident infection reported a significantly higher number of contacts outside of the household compared with participants without incident infection (median: 5 versus 3, p = 0.005). The incidence of infection was significantly higher among unvaccinated/partially vaccinated participants than among fully vaccinated participants (9/7,679 vs. 0/6,845 cases per person-week; Fisher's exact: p = 0.004).

Seroconversion (anti-Spike IgG) was observed in all eight cases with available follow-up serology, but in none of six unvaccinated individuals retested at the end of the study.

## Discussion

Our study of weekly SARS2 rtPCR surveillance of nasal specimens from adults and children over a period of one year spanning pre- and post-vaccine eras has several notable findings: (1) Weekly SARS2 surveillance by rtPCR identified no asymptomatic and only one pre-symptomatic SARS2 infections; (2) Social distancing from persons outside the household was associated with protection from SARS2 infection; (3) SARS2 vaccination was associated with fewer infections.

Surveillance for SARS2 has been recommended as a modality to limit transmission of infection from unvaccinated individuals with asymptomatic or pre-symptomatic infection [3]. While acceptance of long-term surveillance testing was extremely high among our study population as demonstrated by high adherence to collection and return of swabs, the utility of weekly surveillance by rtPCR in our population was of limited value because of our extremely low rate of detecting asymptomatic or pre-symptomatic infection in unvaccinated participants. Only one of nine cases was detected by our SARS2 surveillance when pre-symptomatic and all others experienced COVID-19 symptoms prior to the time when swabs were collected. The absence of asymptomatic infections in our participants may reflect their age and social distancing practices, whereas prior studies show the greatest proportion of asymptomatic infections among children [1]. More frequent testing and more rapid return of results may have increased the detection of pre-symptomatic infection, as reported by others [6, 7] and by modeling of SARS2 surveillance strategies [8].

Acquisition of SARS2 infection in our participants was associated with a higher number of contacts outside of their household. The limited number of social contacts in our study population suggests a high awareness of viral transmission and the ability to implement social distancing practices, including by the many participants who continued to work at their pre-pandemic worksites. Participants with incident infection reported additional discrete risk factors such as being a healthcare worker, having close contact with a healthcare worker, attending in-person school, or having a young child in childcare. These observations are consistent with higher case detection by surveillance rtPCR in unvaccinated health-care workers [9] and in urban public schools in Omaha, Nebraska [10]. Given these observations, risk factor-based screening or viral surveillance of populations known to be at high risk for SARS2 may be of greater value than viral surveillance of populations who are able to practice social distancing.

Our cohort was fully enrolled prior to the availability of the SARS2 vaccine, and the study extended across the winter 2020–2021 surge in cases when the regional 7-day incidence was more than 200 cases per 100,000 population [11]. More than half of the observed person time occurred in unvaccinated or partially vaccinated individuals. When SARS2 vaccines became available in January 2021, our participants demonstrated a relatively high and rapid uptake of vaccination. All individuals with incident infections were unvaccinated, except for one partially vaccinated participant who likely contracted the infection ~2 weeks following their first vaccine. The strong association of incident SARS2 infections with unvaccinated status during the time of the 2021 winter surge in infections, reflects a strong protective effect from vaccination, as has been observed in clinical trials [12, 13].

Limitations of our study include that our participants' age, health, and education, as well as a high awareness of primary SARS2 prevention strategies, may not extend to other at-risk populations. Other communities with less opportunity to practice primary prevention measures may detect a greater rate of SARS2 infections by weekly surveillance strategies [10], and newer

sensitive technologies, such as loop-mediated isothermal amplification, with decreased cost and turn-around time may increase the value of asymptomatic screening more broadly [14]. Second, due to employment by a health care facility, many of our study participants had access to and uptake of SARS2 vaccines shortly after EUA during the 2021 winter surge. Recent vaccination may have exaggerated the significant association between incident infection and lack of full vaccination (i.e., no breakthrough cases were identified secondary to waning immunity) [15]. Third, our observation period, which began early in the pandemic, was prior to the emergence of the more transmissible Delta and Omicron variants which demonstrate vaccine escape compared to prior variants [16, 17]. In communities with high case rates or highly transmissible variants with potential for vaccine escape, prospective rtPCR screening of asymptomatic individuals may be of higher value. Fourth, we sampled participants only once weekly and may have missed asymptomatic infections [6, 7], although end of study antibody testing in unvaccinated individuals did not reveal any additional infections. Finally, our study utilized self-report of symptoms, number of close contacts, and exposure to known cases and was conducted in a place of employment, which may have led to reporting bias.

In conclusion, prospective surveillance for SARS2 infections by weekly PCR screening of asymptomatic unvaccinated individuals may be low yield among individuals who practice primary prevention strategies, particularly in those without workplace or household exposures to SARS2.

## Supporting information

**S1 Table. Primary data.**
(XLSX)

## Acknowledgments

We would like to acknowledge the individuals who participated in the SARS2 Study.

## Author Contributions

**Conceptualization:** Whitney E. Harrington, John D. Aitchison, Lisa M. Frenkel.

**Data curation:** Whitney E. Harrington, Winnie Yeung, John Houck, Sheila Styrchak, Micaela Haglund, Tina Nguyen.

**Formal analysis:** Whitney E. Harrington, Lisa M. Frenkel.

**Funding acquisition:** Whitney E. Harrington, John D. Aitchison, Lisa M. Frenkel.

**Investigation:** Ingrid A. Beck, Fred D. Mast, Sheila Styrchak, Leslie R. Miller, Song Li, Yonghou Jiang, Blair Armistead, Daisy Ko, Alyssa Oldroyd, Ana Gervassi.

**Methodology:** Ingrid A. Beck, Fred D. Mast, Sheila Styrchak, Leslie R. Miller, Ana Gervassi.

**Project administration:** Winnie Yeung, Micaela Haglund, Jackson Wallner, Tina Nguyen.

**Resources:** Whitney E. Harrington, John D. Aitchison, Lisa M. Frenkel.

**Supervision:** Whitney E. Harrington, John D. Aitchison, Lisa M. Frenkel.

**Visualization:** Whitney E. Harrington.

**Writing – original draft:** Whitney E. Harrington, Lisa M. Frenkel.

**Writing – review & editing:** Winnie Yeung, Ingrid A. Beck, Fred D. Mast, John Houck, Sheila Styrchak, Leslie R. Miller, Song Li, Micaela Haglund, Yonghou Jiang, Blair Armistead,

Jackson Wallner, Tina Nguyen, Daisy Ko, Samantha Hardy, Alyssa Oldroyd, Ana Gervassi, John D. Aitchison, Lisa M. Frenkel.

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
