## [Decision Letter · Decision Letter 0]

29 Jul 2022

PONE-D-22-19281Low rate of SARS-CoV-2 incident infection identified by weekly screening PCR in a prospective year-long cohort studyPLOS ONE

Dear Dr. Frenkel,

Thank you for submitting your manuscript to PLOS ONE. After careful consideration, we feel that it has merit but does not fully meet PLOS ONE’s publication criteria as it currently stands. Therefore, we invite you to submit a revised version of the manuscript that addresses the points raised during the review process.

ACADEMIC EDITOR: As appended below, the reviewers have raised some concerns/critiques and suggested further justification/work to consolidate the findings. Do go through the comments and amend the MS accordingly. ==============================

We look forward to receiving your revised manuscript.

Kind regards,

A. M. Abd El-Aty

Academic Editor

PLOS ONE

Journal Requirements:

Reviewers' comments:

Reviewer's Responses to Questions

**Comments to the Author**

1. Is the manuscript technically sound, and do the data support the conclusions?

Reviewer #1: Yes

Reviewer #2: Yes

Reviewer #3: Yes

2. Has the statistical analysis been performed appropriately and rigorously? 

Reviewer #1: Yes

Reviewer #2: Yes

Reviewer #3: N/A

3. Have the authors made all data underlying the findings in their manuscript fully available?

Reviewer #1: Yes

Reviewer #2: Yes

Reviewer #3: No

4. Is the manuscript presented in an intelligible fashion and written in standard English?

Reviewer #1: Yes

Reviewer #2: Yes

Reviewer #3: Yes

5. Review Comments to the Author

Reviewer #1: The manuscript submitted by Frenkel et al and entitled “Low rate of SARS-CoV-2 incident infection identified by weekly screening PCR in a prospective year-long cohort study” evaluates the utility of asymptomatic screening for SARS-CoV-2 in low-risk populations. In this work 553 seronegative individuals were followed for 1 year and tested using self-collected nasal swabs for rtPCR testing. Of the subjects tested, only 9 cases were identified as positive and included subjects at greater risk, HCP, house member of a HCP, student, and daycare case. Furthermore, due to the once-a-week screening, 1 of 9 were identified prior to symptom onset and self-requested testing. Overall, the study is well written and demonstrates that in low risk, high educated settings, a weekly screening had low impact on identifying pre-symptomatic individuals. A few modifications should be addressed prior to publication.

Major Comments

What was the prevalence in the area during the period of testing and do you have that available for asymp vs symptomatic subjects (greater area vs study population). I would also suggest adding the date range to the materials and methods section.

Minor Comments

Ln 66: Why was transport media not used for the study and was there any stability testing done in house to demonstrate that dry swabs remained detectable over the 72 hours.

Figure 1: What do the 2 no stars represent? Is that no specific variable determined?

Conclusion: Add a limitation that contacts are all self-reported and as this was work done at a place of employment, individuals may have misrepresented their social interactions.

Reviewer #2: This is a very interesting study providing intriguing evidence for the usefulness of the weekly testing with rtPCR. RTPCR has its limits due to the price and the need to purify the samples. In one recent study (PMID: 35395856), completely alternative approach was suggested with much faster turnaround time without compromising the quality, sensitivity and specificity. Maybe authors can elaborate the discussion about this point a bit more. Would the faster and cheaper testing method improve the proposed population surveillance outcomes?

Reviewer #3: This is a very nice paper.

Short Title: I think that shortening SARS-CoV-2 to SARS2 is unwise. Perhaps “rate of incident” can be changed to “incidence” to accommodate more complete nomenclature.

Line 76: How was the sensitivity of detection for the nasal specimens determined?

I think that it should be possible to address the HIPAA considerations in creating a deidentified public data set by replacing dates with the intervals between dates, and use of age groupings rather than age. This should represent sufficient deidentification. The questionairre data obviously constitutes identification, but the details are unimportant for purposes of this paper.

6. PLOS authors have the option to publish the peer review history of their article (what does this mean?). If published, this will include your full peer review and any attached files.

Reviewer #1: No

Reviewer #2: No

Reviewer #3: No

---

## [Author Response · Author response to Decision Letter 0]

8 Aug 2022

Reviewers' Comments:

Reviewer #1: 

The manuscript submitted by Frenkel et al and entitled “Low rate of SARS-CoV-2 incident infection identified by weekly screening PCR in a prospective year-long cohort study” evaluates the utility of asymptomatic screening for SARS-CoV-2 in low-risk populations. In this work 553 seronegative individuals were followed for 1 year and tested using self-collected nasal swabs for rtPCR testing. Of the subjects tested, only 9 cases were identified as positive and included subjects at greater risk, HCP, house member of a HCP, student, and daycare case. Furthermore, due to the once-a-week screening, 1 of 9 were identified prior to symptom onset and self-requested testing. Overall, the study is well written and demonstrates that in low risk, high educated settings, a weekly screening had low impact on identifying pre-symptomatic individuals. A few modifications should be addressed prior to publication.

Major Comments

What was the prevalence in the area during the period of testing and do you have that available for asymp vs symptomatic subjects (greater area vs study population). 

-We thank the reviewer for their feedback. The prevalence of symptomatic infection in the greater area (King County) varied across the year but reached a peak 7-day incidence of more than 200 cases / 100,000 population. This information is presented in the fourth paragraph of the discussion, along with a citation to the King County Department of Health website with full rolling incidence of infection throughout the pandemic. Unfortunately, to the best of our knowledge there were no large studies of asymptomatic individuals from the larger population during the time period of our study, so this information is not available for comparison. 

I would also suggest adding the date range to the materials and methods section.

-We have moved the information on date range of enrollment from the first sentence of the Results to the first sentence of the Material and Methods. 

Minor Comments

Ln 66: Why was transport media not used for the study and was there any stability testing done in house to demonstrate that dry swabs remained detectable over the 72 hours.

-At the time of the study there were transport media shortages globally, so we conducted in house testing to demonstrate that dry swabs retained similar sensitivity up to 72 hours. Specifically, we compared the cycle threshold (Ct) values of nasal swabs spiked with 50ul of a SARS-CoV-2 positive specimen collected in universal transport medium (UTM), when held in UTM, PBS, or dry, and found that the cycle time remained stable over 72 hours regardless of storage condition. A comment noting this is now included in the Materials & Methods. 

Figure 1: What do the 2 no stars represent? Is that no specific variable determined?

-We have clarified in the figure legend that “no stars” represents the two cases with no risk factor for infection identified.

Conclusion: Add a limitation that contacts are all self-reported and as this was work done at a place of employment, individuals may have misrepresented their social interactions.

-We have added these limitations to the conclusions.

Reviewer #2: 

This is a very interesting study providing intriguing evidence for the usefulness of the weekly testing with rtPCR. RTPCR has its limits due to the price and the need to purify the samples. In one recent study (PMID: 35395856), completely alternative approach was suggested with much faster turnaround time without compromising the quality, sensitivity and specificity. Maybe authors can elaborate the discussion about this point a bit more. Would the faster and cheaper testing method improve the proposed population surveillance outcomes?

-We appreciate the reviewer calling this publication to our attention. We have added a brief discussion of how alternative testing approaches may change the value of surveillance to the manuscript, citing this publication.

Reviewer #3: This is a very nice paper.

Short Title: I think that shortening SARS-CoV-2 to SARS2 is unwise. Perhaps “rate of incident” can be changed to “incidence” to accommodate more complete nomenclature.

-We appreciate this feedback and have edited the Short Title to: “Low incidence of SARS-CoV-2 by weekly PCR”

Line 76: How was the sensitivity of detection for the nasal specimens determined?

-The sensitivity of detection was determined using 2-fold serial dilutions of an in-vitro transcribed SARS-CoV-2 RNA control spiked into SARS-CoV-2 negative nasal swab matrix (nasal swabs collected from healthy donors). The analytical sensitivity of the diagnostic RT-qPCR was estimated at 10 SARS-CoV-2 RNA copies/reaction (equivalent to 250 copies of SARS-CoV-2 per nasal swab eluted in 1 mL of PBS) utilizing a cycle threshold cut-off of 40. To confirm the limit of detection (LoD) we tested 20 replicate samples at 1x and 20 replicates at 2x the estimated LoD, resulting in 95% positive replicates at 250 copies/swab and 100% positivity at 500 copies/swab. We have included a brief description of our approach in the third paragraph of the Materials & Methods. 

I think that it should be possible to address the HIPAA considerations in creating a deidentified public data set by replacing dates with the intervals between dates, and use of age groupings rather than age. This should represent sufficient deidentification. The questionnaire data obviously constitutes identification, but the details are unimportant for purposes of this paper.

-We appreciate this suggestion and have generated a primary data set with age blocks as suggested and vaccine dates removed, but compiled number of vaccinated individuals per week of the study presented.

---

## [Decision Letter · Decision Letter 1]

23 Aug 2022

Low rate of SARS-CoV-2 incident infection identified by weekly screening PCR in a prospective year-long cohort study

PONE-D-22-19281R1

Dear Dr. Frenkel,

We’re pleased to inform you that your manuscript has been judged scientifically suitable for publication and will be formally accepted for publication once it meets all outstanding technical requirements.

Kind regards,

A. M. Abd El-Aty

Academic Editor

PLOS ONE

Additional Editor Comments (optional):

Reviewers' comments:

Reviewer's Responses to Questions

**Comments to the Author**

1. If the authors have adequately addressed your comments raised in a previous round of review and you feel that this manuscript is now acceptable for publication, you may indicate that here to bypass the “Comments to the Author” section, enter your conflict of interest statement in the “Confidential to Editor” section, and submit your "Accept" recommendation.

Reviewer #2: All comments have been addressed

Reviewer #3: All comments have been addressed

2. Is the manuscript technically sound, and do the data support the conclusions?

Reviewer #2: Yes

Reviewer #3: (No Response)

3. Has the statistical analysis been performed appropriately and rigorously? 

Reviewer #2: Yes

Reviewer #3: (No Response)

4. Have the authors made all data underlying the findings in their manuscript fully available?

Reviewer #2: Yes

Reviewer #3: (No Response)

5. Is the manuscript presented in an intelligible fashion and written in standard English?

Reviewer #2: Yes

Reviewer #3: (No Response)

6. Review Comments to the Author

Reviewer #2: (No Response)

Reviewer #3: (No Response)

7. PLOS authors have the option to publish the peer review history of their article (what does this mean?). If published, this will include your full peer review and any attached files.

Reviewer #2: No

Reviewer #3: No

---

## [Editor Report · Acceptance letter]

15 Sep 2022

PONE-D-22-19281R1 

Low rate of SARS-CoV-2 incident infection identified by weekly screening PCR in a prospective year-long cohort study 

Dear Dr. Frenkel:

I'm pleased to inform you that your manuscript has been deemed suitable for publication in PLOS ONE. Congratulations! Your manuscript is now with our production department. 

Kind regards, 

on behalf of

Prof. A. M. Abd El-Aty 

Academic Editor

PLOS ONE